# Local Anesthetic Systemic Toxicity Following Inadvertent Intravenous Levobupivacaine Infusion in Infants: A Case Report

**DOI:** 10.3390/medicina59050981

**Published:** 2023-05-19

**Authors:** Justina Jermolajevaite, Ilona Razlevice, Vaidotas Gurskis, Dovile Evalda Grinkeviciute, Laura Lukosiene, Andrius Macas

**Affiliations:** 1Department of Anesthesiology, Medical Faculty, Medical Academy, Lithuanian University of Health Sciences, 44307 Kaunas, Lithuania; ilona.razlevice@lsmu.lt (I.R.); laura.lukosiene@lsmu.lt (L.L.); andrius.macas@lsmu.lt (A.M.); 2Department of Pediatrics, Medical Faculty, Medical Academy, Lithuanian University of Health Sciences, 44307 Kaunas, Lithuania; vaidotas.gurskis@lsmu.lt (V.G.); dovileevalda.grinkeviciute@lsmu.lt (D.E.G.)

**Keywords:** local anesthetic systemic toxicity, anesthesia adverse event, pediatric

## Abstract

*Background and objectives:* Local anesthetic systemic toxicity (LAST) in children is extremely rare, occurring at an estimated rate of 0.76 cases per 10,000 procedures. However, among reported cases of LAST in the pediatric population, infants and neonates represent approximately 54% of reported LAST cases. We aim to present and discuss the clinical case of LAST with full clinical recovery due to accidental levobupivacaine intravenous infusion in a healthy 1.5-month-old patient, resulting in cardiac arrest necessitating resuscitation. *Case presentation:* A 4-kilogram, 1.5-month-old female infant, ASA I, presented to the hospital for elective herniorrhaphy surgery. Combined anesthesia was planned, involving general endotracheal and caudal anesthesia. After anesthesia induction, cardiovascular collapse was noticed, resulting in bradycardia and later cardiac arrest with EMD (Electromechanical Dissociation). It was noticed that during induction, levobupivacaine was accidentally infused intravenously. A local anesthetic was prepared for caudal anesthesia. LET (lipid emulsion therapy) was started immediately. Cardiopulmonary resuscitation was carried out according to the EMD algorithm, which lasted 12 min until spontaneous circulation was confirmed and the patient was transferred to the ICU. In ICU, the girl was extubated the second day, and the third day she was transferred to the regular pediatric unit. Finally, the patient was discharged home after a total of five days of hospitalization with full clinical recovery. A four-week follow-up has revealed that the patient recovered without any neurological or cardiac sequelae. *Conclusions:* The clinical presentation of LAST in children usually begins with cardiovascular symptoms because pediatric patients are already under general anesthesia when anesthetics are being used, as was the case in our case. Treatment and management of LAST involve cessation of local anesthetic infusion, stabilization of the airway, breathing, and hemodynamics, as well as lipid emulsion therapy. Early recognition of LAST as well as immediate CPR if needed and targeted treatment for LAST can lead to good outcomes.

## 1. Introduction

Local anesthetic systemic toxicity (LAST) in children is extremely rare, occurring at an estimated rate of 0.76 cases per 10,000 procedures [1]. However, among reported cases of LAST in the pediatric population, infants and neonates represent approximately 54% of reported cases [2]. Systemic absorption of local anesthetic can occur during bolus infusion or inadvertent intravenous injection, resulting in neurological and cardiovascular complications and death. Methemoglobinemia can also be present in some cases of LAST [3,4].

The mechanism of action of local anesthetics is mainly described by the blockage of voltage-gated Na^+^ channels, which leads to the prevention of further nerve impulse conduction in the membrane. The therapeutic goal of this mechanism is usually to block the nerve tissues involved in pain transmission; however, if anesthetic enters systemic circulation, local anesthetic blockage has an effect on any tissue containing sodium channels, including cardiac and CNS tissues, which leads to these systems’ functional derangement [5,6].

We present a case report that demonstrates LAST with full clinical recovery due to an accidental levobupivacaine intravenous infusion in a healthy 1.5-month-old patient, resulting in cardiac arrest necessitating resuscitation. The described occurrence is unexpected and novel since there have not been previously published accidental local anesthetic intravenous injections in infants.

## 2. Case Presentation

A 4-kg, 1.5-month-old female, ASA I, without any medical history, presented for elective one-sided hernia repair. A review of the systems did not reveal any problems. Combined anesthesia was planned, involving general endotracheal and caudal anesthesia. The infant presented to the operating room in a calm, nonagitated state with a heart rate (HR) of 135 bpm and a blood pressure (BP) of 105/60 mmHg.

The infant underwent sevoflurane induction with mivacurium (0.5 mg) and dexamethasone (0.5 mg) and was intubated. Immediately after that, bradycardia at 90 bpm occurred. Atropine (20 μg/kg) was administered two times, as was epinephrine (10 μg/kg), without any effect. Displacement of the endotracheal tube and pneumothorax were excluded by patient lung auscultation; moreover, all artificial lung ventilation parameters were in normal ranges according to patient age and weight. Cardiovascular diseases were considered; however, previous medical history and the patient’s clinical condition before anesthesia did not demonstrate any symptoms or signs that could indicate congenital cardiovascular abnormalities. The nurse confessed that during induction, levobupivacaine was accidentally infused intravenously. The mentioned medication was prepared for caudal anesthesia. Presently, intralipid infusion of 1.5 mL/kg bolus was started with continuation of 0.25 mL/kg/min and administration of epinephrine 10 μg/kg to treat bradycardia. The call for a resuscitation team was made. The bone marrows were punctured, and intravenous dobutamine administration was started following a NaHCO_3_ 4.2% 4 mL infusion because, in the arterial blood gas test, severe acidosis was observed (pH 6.9, base excess −15.9 mEq/L). In the cardiac monitor, wide QRS complexes were seen and no pulse was identified; consequently, electromechanical dissociation (EMD) was confirmed, and cardio-pulmonary resuscitation (CPR), including chest compression and epinephrine administration, was started according to the algorithm. CPR continued for 12 min, and i/v epinephrine was repeated three times. Finally, the recovery of spontaneous circulation was confirmed. After resuscitation, HR was 152 bpm, and BP was 97/42 mmHg with continued dobutamine infusion. While continuing artificial lung ventilation, the observed SpO_2_ value was 98%, and the patient was transferred to the pediatric ICU with continuous intralipid and dobutamine infusions (Figure 1).

Intralipid was stopped after reaching a total infusion of 60 mL. Vasopressors were discontinued the next day. A transthoracic echocardiogram (TTE) was performed the following day, showing a structurally normal heart. The second day, the patient was extubated, and the third day, she was transferred to a regular pediatric unit. Finally, the patient was discharged home after a total of five days of hospitalization. Later on, any blood test abnormalities, including methemoglobinemia, were not observed.

A four-week follow-up has revealed that the patient recovered without any neurological or cardiac sequelae. Consequently, elective surgery was performed without any adverse events. Further neurological assessment of the patient mentioned is planned within the next two years.

## 3. Discussion

Local anesthetic systemic toxicity (LAST) in the pediatric population is an understudied complication of regional anesthesia, but it may differ in some aspects when compared with adults’ LAST. The incidence of LAST in children is counted at around 8/100,000 procedures [1], whereas the risk in adults is much higher at 27/100,000 [5]. A recent literature review performed by Ramesh and Boretsky found 31 case reports of LAST among children reported from 2014 to 2019. Surprisingly, no deaths or long-term morbidity have been reported among them, contrary to adults’ mortality rate of LAST, which varies between 4.3% and 10% [7,8].

The clinical presentation of LAST can vary, but mostly it involves CNS and cardiovascular symptoms. Even though CNS signs and symptoms mostly present first, children can develop cardiotoxicity prior to neurological manifestation [3,9]. This is explained by the fact that usually pediatric patients are already under general anesthesia when local anesthetics are administered, so infants are unable to communicate changes in the CNS such as dizziness, tinnitus, taste alterations, paresthesia, tremors, etc. Cardiovascular manifestations of LAST involve arrhythmias and myocardial depression. The range of arrhythmias is wide and can vary from bradyarrhythmias to reentry tachyarrhythmias and wide-complex arrhythmias. If a large dose of local anesthetic is injected into the systemic circulation, it can provoke immediate cardiac arrest [6]. Early symptoms are not specific only to LAST; therefore, caregivers administering or preparing local anesthetics for a particular patient should always keep in mind the possibility of LAST.

In our case, the patient first presented bradycardia, which was refractory to atropine and epinephrine, and later it progressed to wide QRS complexes with no pulse-EMD with the necessity of resuscitation, which lasted 12 min. It is important to note that the development of acute hemodynamic depression in infants after induction of anesthesia, as in our case, has a broad differential diagnosis, including ETT displacement, myocardial depression caused by anesthetics, pneumothorax, undiagnosed congenital diseases, and, finally, LAST. Therefore, to identify this complication requires vigilance and clinical susceptibility.

Management and treatment of LAST start with early identification of the signs and symptoms noted before stabilizing the airway, breathing, and circulation. The cessation of local anesthetic infusions should take place immediately. Secondly, airway security to assure oxygenation and ventilation should be made in order to prevent hypoxia, hypercarbia, or acidosis. Our patient was already intubated, so we just checked the ETT position. Treatment depends on present symptoms—if cardiovascular changes are present, such as bradycardia, as it was in our case—and the administration of atropine 20 μg/kg (repeated once) and/or epinephrine 10 μg/kg (repeated every 3–5 min) according to PALS guidelines (Pediatric Advances Life Support) [10], although according to ASRA guidelines on LAST (American Society of Regional Anesthesia and Pain Medicine), they suggest a lower dose of epinephrine by 1 μg/kg, but there is no explanation why they reduce this dose [5]. Speaking of present CNS symptoms, such as convulsions (which happen less frequently in the pediatric population because they are already under general anesthesia), benzodiazepines should be given. If cardiac arrest is present, CPR with effective chest compression and administration of epinephrine (10 μg/kg) have to be started according to the PALS algorithm. LET (lipid emulsion therapy) should be started immediately after suspicion of the LAST event. Although the specific ILE’s mechanism of action is unclear, some authors explain the theory of “lipid sink”. A LA concentration gradient is formed between the tissues and the blood, due to which the LA are removed from the heart or brain tissue, where there are high concentrations of them, into the “lipid sink”. In this way, the amount of free LA decreases and causes less harm to organs sensitive to the effects of local anesthetics [5]. Hence, ILE binds LA molecules and increases the volume of distribution of LA, thereby reducing the unwanted effects of LA. The recommendations for dosage remain the same for adults as for children, starting with a 1.5 mL/kg bolus over 2–3 min with continuous infusion of 0.25 mL/kg/min and not exceeding the maximum lipid dose of 12 mL/kg [5]. If cardiovascular stability is reached, LET should be continued for 10 min. In our case, we started lipid therapy as soon as a probable diagnosis of LAST was made, approximately 2–3 min after accidental infusion, with recommended doses, but the patient remained hemodynamically unstable and was transferred to the ICU with continuous infusion of LET and dobutamine. The total dose of infused lipid was 60 mL, which means 15 mL/kg. However, it did not contribute to the patients’ outcomes. Furthermore, it is noted that some patients in LAST can develop methemoglobinemia, which is reported only with four types of anesthetics: prilocaine, benzocaine, lidocaine, and tetracaine. However, in laboratory findings, methemoglobinemia can occur due to the infusion of lipid emulsion, which disrupts blood analysis [4].

The presented case report illustrates LAST event provoked by an accidental levobupivacaine direct intravenous infusion in a 1.5-month-old infant with full clinical recovery. Even though LAST manifested with cardiovascular depression resulting in bradycardia and cardiovascular arrest necessitating resuscitation, early recognition of LAST as well as immediate CPR and targeted treatment for LAST led to good outcomes. The mentioned situation was challenging as the infant deteriorated very fast due to LA intravenous injection. When medical staff faces a similar situation during central or peripheral nerve block performance, the LAST can be recognized without delay.

We performed a PubMed search of other reported pediatric LAST cases and case series between 2016 and 2022 and found a total of 16 published cases; most of them were published in the US, with a few in Europe and Asia. Most of the described LAST cases happened in the operating room and mostly involved cardiovascular signs and symptoms. Five out of sixteen cases included cardiac arrest with CPR. Although no sequels due to LAST were mentioned in any case (Table 1).

We want to emphasize not only the importance of strategies to reduce LAST but also the high importance of implementing effective preventive strategies to improve medication errors.

## 4. Conclusions

Local anesthetic systemic toxicity is a life-threatening complication that in children usually begins with cardiovascular symptoms because the pediatric patient is already under general anesthesia when anesthetics are being used, as it was in our case. Treatment and management of LAST involve stabilization of the airway, breathing, and hemodynamics, as well as LET therapy. Early recognition of LAST as well as immediate CPR if needed and targeted treatment for LAST can lead to good outcomes. A cornerstone of the presented clinical case is continuous improvement based on learning from errors and adverse events.

## Figures and Tables

**Figure 1 medicina-59-00981-f001:**
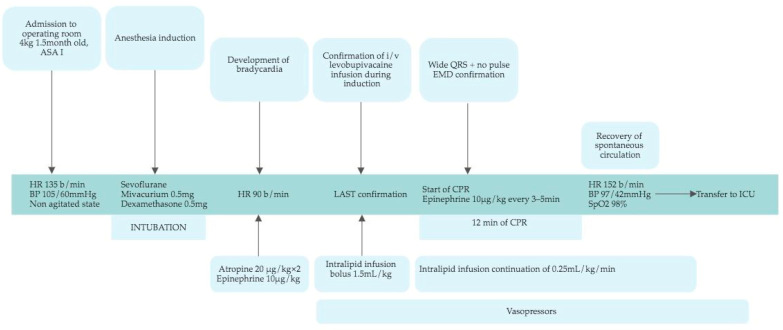
Schematic representation of the case from admission to the operating room to the ICU. HR—heart rate; BP—blood pressure; LAST—local anesthetic systemic toxicity; i/v—intravenously; EMD—electromechanical dissociation; CPR—cardiopulmonary resuscitation; ICU—intensive care unit.

**Table 1 medicina-59-00981-t001:** Summary of published pediatric LAST cases between 2016 and 2022.

No.	Author	Country	Patient’s Age	Type of Surgery	Place of LAST Event	Intended Site of Administration	Accidental i/v Injection	Local Anesthetic	Cardiovascular	CNS	CPR	Medications Used	Outcome
1	Najafi N., Veyckemans et al. [11]	USA	4.5 years	Circumcision	Operating room	Penile block		Ropivacaine	Bradycardia	-	-	Atropine	No sequels
2	Yu R.N., Houck C.S., Casta A., et al. [12]	USA	9 months	Circumcision	Operating room	Penile block		Bupivacaine	ECG changes: wide QRS and peaked T waves	-			No sequels
3	Yu R.N., Houck C.S., Casta A., et al. [12]	USA	3 years and 2 months	Revision circumcision	Operating room	Penile block		Bupivacaine	ECG changes: ST-segment depression and inverted T waves	-			No sequels
4	Yu R.N., Houck C.S., Casta A., et al. [12]	USA	14 months	Distal hypospadias repair	Operating room	Penile block		Bupivacaine	Hypotension ECG changes: wide QRS and peaked T waves	-			No sequels
5	Yu R.N., Houck C.S., Casta A., et al. [12]	USA	9 months	Revision circumcision	Operating room	Penile block		Bupivacaine	Hypotension and atrial flutter	-		Adenosine	No sequels
6	Yu R.N., Houck C.S., Casta A., et al. [12]	USA	10 months	Distal hypospadias repair	Operating room	Penile block		Bupivacaine	Hypotension ECG changes: ST-segment depression and QRS widening	-			No sequels
7	Yu R.N., Houck C.S., Casta A., et al. [12]	USA	6 months	Circumcision	Operating room	Penile block		Bupivacaine	Hypotension ECG changes: wide QRS and peaked T waves	-	4 min CPR	Intralipid	No sequels
8	Yu R.N., Houck C.S., Casta A., et al. [12]	USA	6 months	Circumcision	Operating room	Penile block		Bupivacaine	Hypotension ECG changes: QRS widening and peaked T waves	-	3–4 min CPR	Intralipid	No sequels
9	Hernandez M.A., Boretsky et al. [13]	USA	9 months	Pyeloplasty	Intensive care unit	Paravertebral catheter	Probable	Chloroprocaine	Hypotension	Alerted consciousness and seizures			No sequels
10	Musielak M. and McCall J. [14]	USA	6 years	Skin autography due to burn	Operating room	Local infiltration	Probable	Bupivacaine	Bradycardia, asystole, and hypotension		30 min CPR	Intralipid and vasopressors	No sequels
11	Shapiro P. and Schroeck H. [15]	USA	12 months	Repair of the omphalocele and abdominal surgery	Intensive care unit	Epidural catheter 3rd day		Bupivacaine		Agitation and seizures		Lorazepam	No sequels
12	Yamane Y. and Kagawa T. [16]	Japan	6 years	Thoracotomy and pacemaker implantation	Operating room	Paravertebral catheter		Ropivacaine	Asystole		2 min CPR	Adrenaline, intralipid, and vasopressors	No sequels
13	Torres L.M., Figueroa et al. [17]	Brazil	6 years	Hypospadias repair	Operating room	Caudal block		Bupivacaine	Ventricular tachycardia 3 x EMD		1 min 4 min 9 min CPR	Amiodarone, intralipid, and adrenaline	No sequels
14	Eizaga Rebollar R., García Palacios et al. [18]	Spain	17 months	Colostomy close and bowel anastomose	Operating room	Caudal block		Levobupivacaine	ECG changes: ST severe depression and T increase			Intralipid	No sequels
15	Hsieh X.-X., Hsu et al. [19]	Taiwan	10 years	Tenectomy	Operating room	Intravenously		Lidocaine		Tonic-clonic seizures			Finding a diagnosis of epilepsy
16	McMahon K., Paster et al. [20]	USA	15 months		Emergency room	Topical		Lidocaine and prilocaine cream		Tonic-clonic seizures		Lorazepam and intralipid	No sequels

## Data Availability

The data presented in the present study is available from the corresponding author.

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
