# Peer review of "Local Anesthetic Systemic Toxicity Following Inadvertent Intravenous Levobupivacaine Infusion in Infants: A Case Report"

_medicina, 2023, doi:10.3390/medicina59050981_

Round 1

Reviewer 1 Report

The study case presents one of the rarest cases occurring in children, i.e., LAST caused by the infused levobupivacaine. The case report must be very helpful for the clinicians regarding the medications administered to the infant.

----------------------------

Additional comments:

The manuscript is well written and of interest for the readers. As it is a case report that shares an unique clinical experience on LAST infants, no original methodologies, research questions or figures from data analyses were expected. However, here are some comment to help the authors to improve the manuscript.

1. Is there any innovation step in the procedures and management of LAST for your case, or only the common treatment was applied?

2. The authors state that early recognition of LAST leads to good outcomes. In your case, it was LAST was assessed by the nursery confession, but what are the early CNS and cardio-vascular symptoms of LAST, so other doctor can recognize? Please give more details.

3. As the authors note, the development of acute hemodynamic depression in infant after induction of anesthesia has a broad differential diagnosis. I assume it is important from the treatment viewpoint. Hence how the treatment of LAST differ from ETT displacement, myocardial depression, pneumothorax, congenital diseases? Does applying other specific treatment for the aforementioned cause would deteriorate the LAST-caused state of the infant?

4. What does it add to the subject area compared with other published material?

Author Response

Dear Reviewer,

Thank you very much for Your interest to our article. We want to provide the responses to your the comments and aswell we add the corrected manuscript below. 

Response to Reviewer 1 Comments

  1. Is there any innovation step in the procedures and management of LAST for your case, or only the common treatment was applied?

Response. In presented case the treatment was provided according to adults’ guidelines for management of local anesthetic systemic toxicity adapted to pediatric patients (Dontukurthy S, Tobias JD. Update on Local Anesthetic Toxicity, Prevention and Treatment During Regional Anesthesia in Infants and Children. J Pediatr Pharmacol Ther. 2021;26(5):445-454. doi: 10.5863/1551-6776-26.5.445. Epub 2021 Jun 28. PMID: 34239395; PMCID: PMC8244955.). Symptomatic treatment was provided according to clinical manifestation. Specific antidotes were administered according to LAST treatment recommendations:

  • Sodium bicarbonate for ventricular dysrhythmias. We have prescribed mentioned medication.
  • Methylene blue for methemoglobinemia: 1 - 2 mg/kg IV over 5 minutes. Blood tests were made to investigate methemoglobinemia, but in our case methemoglobin amount was in normal range.
  • Intravenous lipid emulsion in severe cardiovascular toxicity and cardiac arrest was administered immediately to our patient.

  1. The authors state that early recognition of LAST leads to good outcomes. In your case, LAST was assessed by the nursery confession, but what are the early CNS and cardiovascular symptoms of LAST, so other doctor can recognize? Please give more details.

Response. (Discussion section, lines 105-116). CNS symptoms of LAST can vary from lightheadedness, dizziness, tinnitus to coma and respiratory arrest. Pediatric patients in most cases are under general anesthesia or sedation, consequently, cardiovascular symptoms are the primary manifestations of LAST. Early symptoms are not specific only to LAST, therefore, caregivers administering or preparing local anesthetics for particular patient should always keep in mind the possibility of LAST.

  1. As the authors note, the development of acute hemodynamic depression in infant after induction of anesthesia has a broad differential diagnosis. I assume it is important from the treatment viewpoint. Hence how the treatment of LAST differ from ETT displacement, myocardial depression, pneumothorax, congenital diseases? Does applying other specific treatment for the mentioned cause would deteriorate the LAST-caused state of the infant?

Response. (Case Presentation section, lines 63-68). Presented case shows that LAST reducing strategies are crucial and were kept in mind, as follows: adequate oxygenation and ventilation, local anesthetic dosage modification for infant,  using of  local anesthetic agents with lower risk of toxicity etc. After patient deterioration all possible causes and diseases were excluded without delay: ETT displacement and pneumothorax were excluded during auscultation, moreover, all artificial lung ventilation parameters were in normal ranges according to patient age and weight. Cardiovascular diseases were considered, however, previous medical history and patient clinical condition before anesthesia did not demonstrate any symptoms and signs which could indicate congenital cardiovascular abnormalities.

Early identification of LAST is a cornerstrone in saving patient life. Specific treatment for LAST starts immediately after recognition or suspicion of mentioned complication. If patient condition manifests with severe cardiovascular symptoms resuscitation according the Pediatric Advanced Life Support guidelines and lipid emulsion therapy has to be started  immediately. If patient does not response to lipid emulsion therapy within 3 minutes and  repeated  bolus of intralipid are without any response, ECMO has to be considered.

Propofol can worsen cardiac dysfunction that may develop with LAST.

In our case patient outcome would be fatal without specific antidote as the infusion was intravenous and could cause high LA plasma concentration in short time.

  1. What does it add to the subject area compared with other published material?

Response. (Introduction section, lines 51-53). The described occurrence is unexpected and novel since there are not previously published accidental local anesthetic intravenous injections in infants.

(Discussion section, lines 176-178) We want to emphasize not only the importance of strategies to reduce LAST, but also high importance of implementing effective preventive strategies to improve medication errors.

(Conclusions section, lines 186-188) A cornerstone of the presented clinical case is continuous improvement based on learning from errors and adverse events.

Please see the attachment below with corrected manuscript. 

Reviewer 2 Report

I was glad to review the work of the authors regarding this very interesting case report on Local anesthetic systemic toxicity following inadvertent intravenous Levobupivacaine infusion in infant. The manuscript is well-written and the topic is very interesting.

In general, the Manuscript may benefit from several major revisions, as suggested below:

-Local anesthetic systemic toxicity is a life-threatening complication, especially in children. I would suggest adding a figure to make the case report more attractive. It could be a schematic representation of the patient management including medicine, time that was given, dosage, etc. This figure can make the study easier to follow.

- In addition, I would suggest adding a table including similar case reports of Local anesthetic systemic toxicity. There are not many similar cases in the literature. Information that may be included is age, country, medicine that was given, CPR, etc.

Author Response

Dear Reviewer,

Thank you very much for Your interest to our article. We want to provide the responses to your comments and aswell we add the corrected manuscript below. 

Response to Reviewer 2 Comments

  1. Local anesthetic systemic toxicity is a life-threatening complication, especially in children. I would suggest adding a figure to make the case report more attractive. It could be a schematic representation of the patient management including medicine, time that was given, dosage, etc. This figure can make the study easier to follow.

Response. As suggested this data is provided in newly added Figure 1. „Schematic representation of the case from admission to operating room to ICU“.

  1. In addition, I would suggest adding a table including similar case reports of Local anesthetic systemic toxicity. There are not many similar cases in the literature. Information that may be included is age, country, medicine that was given, CPR, etc.

Response. Data is provided in Table 1. “Summary of published pediatric LAST cases between 2016 and 2022”. (Discussion section, lines 167-172). This table is giving information of patients’ age, type of surgery, medicaments given, CNS and cardiovascular symptoms that appeared and CPR (if it was needed to perform). We found total of 16 published cases (between 2016 and 2022), most of them were published in US, few in Europe and Asia. Most of the described LAST cases happened in operating room and mostly involved cardiovascular signs and symptoms. 5 out of 16 cases included cardiac arrest with CPR. Although no sequels due to LAST were mentioned in any case.

Please see the attachment below with corrected manuscript. 

Round 2

Reviewer 2 Report

I would like to congratulate the authors on their fascinating work. All requested changes were addressed accordingly. It can be accepted for publication without further corrections.